# Jujube Powder Enhances Cyclophosphamide Efficiency against Murine Colon Cancer by Enriching CD8^+^ T Cells While Inhibiting Eosinophilia

**DOI:** 10.3390/nu13082700

**Published:** 2021-08-04

**Authors:** Huiren Zhuang, Nan Jing, Luoyang Wang, Guoqiang Jiang, Zheng Liu

**Affiliations:** 1Key Lab of Industrial Biocatalysis, Ministry of Education, Department of Chemical Engineering, Tsinghua University, Beijing 100084, China; zhuanghr19@mails.tsinghua.edu.cn (H.Z.); jn17@mails.tsinghua.edu.cn (N.J.); wangnan235@qdu.edu.cn (L.W.); liuzheng@mail.tsinghua.edu.cn (Z.L.); 2School of Basic Medicine, Qingdao University, Qingdao 266071, China

**Keywords:** chemotherapy, cyclophosphamide, gut microbiota, jujube, murine colon cancer

## Abstract

Cyclophosphamide (CTX) is widely applied in cancer treatment. However, the outcome is often compromised by lymphopenia, myelosuppression, and gut dysbiosis. Here, we used jujube powder to enhance CTX efficiency through nurturing gut microbiota in order to facilitate favorable metabolisms. It was observed that the oral administration of jujube powder enriched CD8^+^ T cells in mouse MC38 colon tumor microenvironment and increased the diversity of gut microbiota and the abundance of *Bifidobacteriales*, which is helpful to the production of butyrate in the cecum content. The application of jujube powder also stimulated the production of white blood cells, especially CD8^+^ T cells in peripheral and bone marrow, while inhibiting the growth of eosinophils in peripheral blood and the production of IL-7 and GM-CSF in serum. All these are conductive to the significant inhibition of the tumor growth, suggesting the high potential of nurturing gut microbiota with natural products for improving the efficiency of chemotherapy.

## 1. Introduction

Cytotoxic chemotherapeutic agents are widely applied in cancer treatment, particularly in conjunction with neoplasm [1,2]. Generally, the efficacy is dose-dependent and often accompanied with adverse effects, such as lymphopenia [3,4], myelosuppression [5], multi-organ failure [6], and gastrointestinal mucositis [7]. The reduction of dose, however, may compromise the effectiveness of chemotherapy [8].

Recent years have witnessed the growing efforts to explore the interaction between gut microbiome and chemotherapy. On the one hand, chemotherapeutic agents alter the abundance and composition of gut microbiota, leading to immune responses or adverse effects [9]. For example, Viaud et al. found that CTX induced the translocation of commensals into secondary lymphoid organs and enhanced therapeutic immunomodulatory effects by increasing memory T helper (Th1) and Th17 responses via NOD_2_ receptors [10]. Stringer et al. reported that the administration of 5-fluorouracil (5-FU) increased *Clostridium* and *Staphylococcus* species while reducing *Lactobacillus* and *Bacteroides*, causing intestinal dysbiosis [11]. On the other hand, commensal bacteria may affect the outcome of cancer therapy through translocation [10], immunomodulation [12], metabolism [13], and enzymatic degradation [14]. Daillere et al. showed that antibiotics treatment eliminating either gram-positive or total bacteria led to the reduced response of Th17 cell and poor outcome in CTX therapy [12]. Shen et al. found that gut microbiota enhanced the development of hyperalgesia induced by oxaliplatin through mediating TLR4 expression on macrophages [15]. Wallace et al. confirmed that oral administration of β-glucuronidase inhibitors protected mice from irinotecan-induced severe diarrhea [13]. Vande Voorde et al. demonstrated that a mycoplasma infection severely reduced the cytostatic activity of gemcitabine by direct deamination of drug at the nucleoside level [14]. All these findings suggest new ways to improve chemotherapy through modulating gut microbiota to enhance immune responses and, in the meantime, reduce unexpected toxicities.

Alexander et al. summarized recent advances in regulating gut microbiota with probiotics, prebiotics, and synbiotics [16,17,18,19] as well as engineering methods utilizing antibiotics, ecology, and synthetic approaches [20,21,22] as ways to enhance anti-tumor efficacy of chemotherapy [9]. Motoori et al. demonstrated that applying synbiotics in esophageal cancer patients receiving neoadjuvant chemotherapy consisting of cisplatin, docetaxel, and 5-FU could relieve lymphopenia and diarrhea as well as the appearance of febrile neutropenia [17]. Wada et al. confirmed that the application of the probiotics *Bifidobacterium* inhibited the infection of fecal micro flora for patients undergoing cancer chemotherapy for pediatric malignancies [23]. Wang et al. found that the enteric microbiota metabolites, the protopanaxadiol group of ginsenosides, enhanced the effectiveness of 5-FU, inhibiting colorectal cancer cell lines [24].

Ziziphus jujube (*Rhamnaceae* family) has been widely applied in traditional Chinese medicine. Jujube contains carbohydrates [25] that can be fermented in the colon to manufacture short-chain fatty acids (SCFAs), which are conductive to the enhanced immune response [26]. Cai et al. reported that two homogenous biological macromolecules extracted from jujube showed immunological and anti-complementary activities through classical pathway [27]. Jing et al. found that jujube improved the response rate of anti-programmed death 1 (PD − L1) against colon tumor through enriching *Lachnospiraceae* in gut microbiota [28].

In the present study, we used jujube powder as a prebiotic to modulate gut microbiota for an enhanced chemotherapy. We examined the effect of jujube powder in the treatment of an MC38 colon tumor model on mice with CTX. We recorded the volume of colon tumor when mice were treated by CTX with or without jujube powder. We analyzed the effect of jujube powder on the composition and richness of gut microbiota and their metabolites, SCFAs. We also determined CD8^+^T cells in peripheral blood, spleen, bone marrow and tumor microenvironment as well as cytokines, such as IL-7 and GM-CSF, in serum. Encouraging results were obtained in the present study, suggesting a high potential of using jujube as a prebiotic to enhance cancer chemotherapy.

## 2. Materials and Methods

### 2.1. Chemicals and Reagents

The jujube was purchased from Ruoqiang County, Xinjiang Province. Jujube powder was prepared by cryogenic grinding at −20 °C for 10 min, yielding jujube powder of 1–10 μm. Cyclophosphamide (CTX) was purchased from Meryer (Shanghai, China) Chemical Technology Co., Ltd.

### 2.2. Mice and Treatment

Male C57BL/6 mice (6 weeks) were purchased from the Vital River Laboratory Animal Technology Co. Ltd. (Beijing, China). Mice were acclimatized to polycarbonate cages under specific pathogen-free (SPF) conditions in the animal care facilities at Tsinghua University for 7 days before animal experiments. Mice were subcutaneously inoculated with MC38 colon tumor cells (about 1 × 10^6^) at the right flank. The volumes of tumor were measured twice a week, and the formula (width^2^ × length)/2 was used to record tumor volume. When tumor volume reached 200 ± 25 mm^3^ on day 7, mice were randomly divided according to tumor size into four groups as follows: control group (CTR group), CTX-treated group (CTX group), CTX- and jujube-treated group (CTX + J group), and jujube-treated group (J group). Each group had 6 mice per one experimental condition except for bioinformatics analysis. The jujube group had 5 mice because the concentration of one mouse’s fecal DNA did not conform to the requirement of 16S rRNA gene sequencing. The above information has been added in the revised version. For CTX groups, mice were given intraperitoneal (i.p.) injection of CTX (80 mg/kg) dissolved in saline at 7–9 days. For CTR group, mice were given saline at 7–9 days. For the group’s oral administration of jujube, mice were admitted jujube powder (800 mg/kg) dissolved in sterile deionized water every day by gavage at 10–21 days. MC38 colon cancer tumor was obtained from American Type Culture Collection (ATCC). MC38 from American Type Culture Collection (ATCC) was cultured in 5% CO_2_ at 37 °C using Dulbecco’s Modified Eagle’s Medium (Invitrogen, Carlsbad, CA, USA) with addition of 10% heat-inactivated fetal bovine serum (HyClone, Logan, UT, USA). Then MC38 cells were digested to single cells by 5 min treatment with 0.5% trypsin (Sigma-Aldrich, St.Louis, MO, USA). The digestion was terminated by adding Dulbecco’s Modified Eagle’s Medium and subjected to 5 min centrifugation at 400× *g*, 4 °C. The concentration was suspended to 1 × 10^7^ /mL in phosphate-buffered saline (PBS) for subcutaneous injection. All animal experimental procedures were performed according with the guidelines of Animal Care and Use Committee of Tsinghua University (No. 20-LZ1#).

### 2.3. DNA Extraction and Bacterial Identification in Stool Samples

Three weeks after tumor inoculation, feces were obtained and subjected to DNA extraction using the QIAamp DNA Stool Mini Kit (Qiagen, Hilden, Germany). The specific experimental methods referred to standard protocols [29]. The reaction steps were as follows: denaturation 3 min at 95 °C, denaturation 30 s at 95 °C (27 cycles), annealing 30 s at 55 °C, elongation 45 s at 72 °C, and extension 10 min at 72 °C. The products of PCR were tested by electrophoresis in 2% agarose. They were purified through the Gel Extraction Kit of AxyPrep DNA (Biosciences, Union City, CA, USA). The products of PCR were quantificationally analysed by the QuantiFluor-ST (Promega, Madison, WI, USA). Then, they were sequenced by the Illumina MiSeq (Majorbio, Shanghai, China) in accordance with the standard protocols. After removing low-quality sequences through FASTP tool, the UPARSE was used to cluster the rest of the high-quality sequences into OTUs at 97% identity (V. 7.1, http://drive5.com/uparse/, accessed on 29 June 2021). The 16S rRNA gene amplicon data and their raw sequence reads are acquirable through the SRA with project ID PRJNA742472.

### 2.4. Bioinformatics Analysis

The alpha-diversity of experimental group was computed via Vegan package and diagramed by the ggplot2 package in R (version R 3.6.1). The principal component analysis, beta diversity of sample-to-sample dissimilarity, was calculated through R package mixOmics. Then, the top 5% OTUs on the horizontal axis, which best represented the first PLS component, were fetched out of PLS-DA result and diagramed via ggplot2 package. Linear discriminant analysis (LDA) effect size (LEfSe) was used to find out the differentiae in community composition between different experimental groups (http://huttenhower.sph.harvard.edu/galaxy, accessed on 29 June 2021). Phylogenetic Investigation of Communities by Reconstruction of Unobserved States (PICRUSt) was employed to analyze the microbial communities’ functional potential. The functions of differentially abundant PICRUSt, which were among the 400 predicted KEGG modules, were further evaluated by R package DESeq2.

### 2.5. Complete Blood Counts

Mice were bled by retro-orbital sinus puncture into heparinized capillary tubes at 5, 10, and 15 days of CTX treatment and analyzed using XN-1000V automated system (Sysmex, Shanghai, China).

### 2.6. Flow Cytometry of Blood, Bone, Spleen, and Tumor Populations

Bone marrow was obtained by irrigating right leg bones. Tumor was digested by collagenase. The spleen cells were acquired by milling and filtrating by 70-μm nylon Cell Strainers (Falcon-Corning, Cornig, New York, NY, USA). Blood, bone marrow, spleen, and tumor cells were stained and performed on LSRFortessa flow cytometer (BD Biosciences, Union City, CA, USA).

Bleeds were performed by retro-orbital sinus puncture into heparinized capillary tubes. Separation of red blood cells from white blood cells was performed using gradient solution consisting of PBS with 8 × 10^−3^ M EDTA (Invitrogen-Life Technologies, Carlsbad, CA, USA) and 2 units/mL heparin sodium in 50:50 mix with 2% dextran (Sigma-Aldrich, St. Louis, MO, USA). Residual red blood cells were lysed by adding red blood cell lysis buffer (Solarbio, Beijing, China).

Bone marrow was harvested by flushing trimmed femora, tibiae, and ilia with approximately 5 mL per bone Hank’s buffered solutions (Gibco-Life Technologies, Waltham, MA, USA) with 2% fecal bovine serum (Sigma-Aldrich, St. Louis, MO, USA) and 1% HEPES buffer (Gibco-Life Technologies, Waltham, MA, USA). Suspension was filtered before further processing through 70-μm nylon cell strainer. Red blood cell lysis was done as for peripheral blood.

Tumor was peeled off and digested by PBS with 0.5 mg/mL collagenase (Sigma-Aldrich, St. Louis, MO, USA) after rocking 2 h on the shaker (37 °C, 120 r/min), suspension was filtered before further processing through 70-μm nylon cell strainer.

Spleen was weighted by electronic analytical balance (Mettler Toledo, Zurich, Switzerland) and milled through 70-μm nylon cell strainer with Hank’s buffered solutions. Red blood cell lysis was done as for peripheral blood.

### 2.7. Staining

Cells were firstly bound to anti-CD16/32 antibody and then stained with antibody mixes (CD45, CD4, CD8a, Ly6C, Ly6G, Cd11b, F4/80, CD19) at concentration 1:200 for each surface marker antibody in Hank’s buffered saline solution for 15 min in dark environment (Cell-markers used for identification of cell population and fluorophore conjugates can been seen in Table 1 and Table 2). All samples were washed with 1 mL Hank’s buffered saline solution and centrifuged for 5 min (400× *g*, 4 °C). The cells were suspended in the Hank’s buffered saline solution and applied to LSRII cell analyzers (BD Biosciences, Union City, CA, USA). The multicolor flow data analysis was performed using FlowJo 10.6.2.

### 2.8. Determination of Cytokines in Serum

Bleeds were performed by retro-orbital sinus puncture into heparinized capillary tubes. Serum was gained by clotting for 30 min before centrifugation at 2000× *g* for 15 min. The concentrations of cytokines (IL-7, IL-3, GM-CSF, M-CSF, TPO) were determined using commercial ELISA kits (Thermo Fisher, Waltham, MA, USA) according to the instructions of ELISA kits strictly.

### 2.9. Quantification of SCFAs

Cecal contents of 0.2 g was mixed in 0.2 mL 50% sulfuric acid and 0.8 mL water and centrifugated at 12,000× *g* (15 min). The supernatant after centrifugation was extracted through the same volume ethyl and was analyzed on gas chromatography using Shimadzu GC-2010 system (Shimadzu, Kyoto, Japan).

### 2.10. Statistical Analysis

All above statistical analyses were carried out on GraphPad Prism 8. The two-tailed Student’s *t*-test was used to compare differences between groups. The measure of one-way analysis of variance (ANOVA) was used to compare differences among all groups. The results were expressed as means ±SE. *p* < 0.05 represented statistically significant difference. *p* < 0.01 indicated high statistically significant difference. * *p* < 0.05, ** *p* < 0.01.

## 3. Results

### 3.1. Jujube Facilitates Tumor-Infiltrating CD8^+^ T Cells and Enhances CTX Efficiency

The mice were fed daily with 800 mg/kg jujube powder dissolved in sterile deionized water after intraperitoneal (i.p.) injection of 80 mg/kg CTX dissolved in sterile saline on days 7–9. As shown in Figure 1A, the tumor volume for the CTX + J group mice is significantly smaller than that for the CTX group. The administration of jujube powder alone had no obvious effect on tumor size. Moreover, the CTX + J group showed a significantly higher infiltration of CD45^+^ cells in tumor microenvironment (Figure 1B). In comparison with the CTX group, the ration of CD8^+^ T cells of total cells in the CTX + J group increased remarkably in the tumor microenvironment (Figure 1C,D). It is particularly noteworthy that the administration of jujube powder relieves spleen edema caused by cyclophosphamide (Appendix A). These encouraging results suggest that the administration of jujube powder is promising to enhance chemotherapy by enriching CD8^+^ cells while inhibiting side effects.

### 3.2. Restoration of Gut Microbiota with Jujube

#### 3.2.1. Jujube Increased Gut Microbiota Diversity

CTX reduced the alpha diversity of gut microbiota compared to CTR group, and jujube could restore the gut microbiota during CTX treatment (Figure 2A). A significant difference in the similarity index (ANOSIM) in bacterial compositions was observed between CTX group and CTX + J group (*p* = 0.006), which was also confirmed by principal component analysis (PCA) (Figure 2B). The partial least squared–discriminative analysis (PLS–DA) model showed a distinguished separation among CTR group, CTX group, and CTX + J group (Figure 2C), which could be well repeated with the top 5% representative OTUs (Figure 2D). As detailed in Figure 2C, the most discriminative taxa leading to the structural changes in the gut microbiota are from Bacteroidota, Firmicutes, and Actinobacteria.

#### 3.2.2. Jujube Powder Altered the Composition of Gut Microbiota

We compared the difference of microbial structures at different taxonomic levels. At the phylum level, the *Firmicutes* to *Bacteroidota* ratio increased slightly in the CTX + J group in comparison with the CTX group (Appendix A). At the order level, the abundance of *Burkholderiales*, which might be conductive to drug resistance [30], increased in the CTX group compared to CTR group, while it was reduced in the CTX + J group (Figure 3A,B). Moreover, the relative abundance of *Bifidobacteriales* was much higher in the CTX + J group than the CTX group (Figure 3A,B). LEfSe analysis showed differential gut microbiota signatures between these four groups. *Bifidobacteriales* were enriched in the CTX + J group, while *Burkholderiales* were enriched in the CTX group, suggesting that gut microbiota of CTX + J group might enhance the immune system through the better utilization of the metabolisms of carbohydrates than CTX group. The Bray–Curtis Anosim analysis demonstrated that there were significant differences in colony distribution among CTX group and CTX + J group (Figure 4A,B).

#### 3.2.3. Function Differences in the Gut Microbiota

We performed the PICRUSt analysis based on 16S rRNA sequencing data. Several major differential KEGG modules (*p* < 0.05) between the CTX and CTX + J groups were identified using DESeq2, among which carbohydrate digestion and absorption was of particular interest (Figure 4A). The KEGG pathway of carbohydrate digestion and absorption demonstrated significantly growth in the CTX + J group compared with the CTX group (Figure 4B), revealing the possible mechanism of modulation of gut microbiota by jujube.

### 3.3. Jujube Powder Increased the Production of SCFAs

As shown in Figure 5A, the total concentration of SCFAs in CTR group decreased after CTX treatment on day 5, 10, and 15, while the administration of jujube powder increased the concentration of SCFAs. Particularly, CTX treatment decreased the concentration of propionate, which was significantly higher by administration of jujube powder in CTX + J group comparing to CTX group (Figure 5B). In addition, the concentration of SCFAs was generally stable for CTX and CTX + J group except the growth of butyrate in CTX + J group on days 5, 10, and 15 (Figure 5C–E).

### 3.4. Jujube Powder Enriched CD8^+^ T Cells but Reduced Eosinophilia

To evaluate the immunosuppression caused by CTX, we analyzed the immune cell levels and composition in the bone marrow, peripheral blood, and spleen. As shown in Figure 6A, the numbers of white blood cells, especially lymphocytes in peripheral blood, declined sharply after CTX treatment, indicating the occurrence of immunosuppressive status. It is noteworthy that a recovery in the number lymphocytes was observed in CTX + J group. We determined eosinophil, CD8^+^, and CD4^+^ T cells in total white blood cells by flow cytometer (Figure 6B–D). As shown in Figure 6B, the jujube powder suppressed eosinophil induced by CTX. Moreover, the proportion of CD4^+^ T cells and CD8^+^ T cells in the CTX + J was higher than that in the CTX group, suggesting enhanced tumor-killing effects. Moreover, CTX reduced CD8^+^ T cells in bone marrow, while jujube had significant effect on recovering the CD8^+^ T cells (Figure 6F). We also determined the hematopoietic growth factor in serum. As shown in Figure 6E, the concentration of IL-7, GM-CSF, and M-CSF significantly increased after CTX treatment and reduced to the ordinary level in the case of CTX + L group. This also indicates a return to homeostatic levels.

## 4. Discussion

The present study confirmed that the intraperitoneal injection of CTX caused both the reduction of CD8^+^ T cells and the increase of eosinophil in peripheral blood. Nurrochmad et al. observed that CD8^+^ T cells in peripheral blood were reduced by CTX treatment (150 and 100 mg/kg body weight on days 1 and 4) [31]. Escudero-Vilaplana reported that CTX caused increase in eosinophils [32]. Wu et al. demonstrated that CTX treatment (140 mg/kg every 6 days) induced reduction of CD8+ T cells in blood while increasing CD8+ T-cell infiltration into tumors [33]. The administration of jujube powder altered the gut microbiota significantly, as evidenced by the recovery of the alpha diversity index in terms of Shannon and Simpson index (Figure 2A), increased CD8^+^ T cells in peripheral blood (Figure 6D) and tumor microenvironment (Figure 1C), inhibited eosinophilia (Figure 6B), and enhanced the effectiveness of CTX on MC38 colon cancer (Figure 1A). We established that IL-7 determined the size and proliferative state of the resting T-cell pool [34].

In our experiment, the CTX group exhibited a higher IL-7 level compared to the CTR group, whereas the CTX + J group showed a regular IL-7 level. Whereas application of jujube powder barely had effect on recovering CD8^+^ T cells of bone marrow, it improved the CD8^+^ T cells in peripheral blood. Effects of jujube powder on hemopoietic progenitor cell growth should be taken into consideration for future efforts in order to analyze the source of CD8^+^ T in peripheral blood.

This work established that jujube enhanced the diversity of *Bifidobacteriales* and reduced the abundance of *Burkholderiales*. *Bifidobacteriales* is helpful for fermenting jujube powder, thus increasing the diversity of gut microbiota through cross-feeding activities [35]. *Bifidobacteriales* is also related to the production of acetate [36], which is essential to butyrate synthesis [37]. The PICRUST analysis demonstrated that the KEGG module of carbohydrate digestion and absorption was higher in the CTX + J group compared to the CTX group, suggesting that the higher levels of SCFAs and butyrate may be attributed to the increased diversity of *Bifidobacteria*. SCFAs were produced by nondigestible fiber fermentation in gut, which not only regulated the balance of host energy metabolism but played a vital role in intestinal and immune homeostasis [38,39]. The SCFA butyrate could directly enhance cytotoxic CD8^+^ T cells via ID2-dependent IL-12 signaling [40]. We found that jujube increased the concentration of SCFA butyrate in cecum during cyclophosphamide treatment, which promoted the percentage of CD8^+^ T cells of total T cells in peripheral blood.

To find out which ingredients of jujube perform the main function, we conducted experiments using major components of jujube, such as jujube polysaccharides, cyclic adenosine monophosphate, aqueous extracts of jujube, and residual components (Appendix A). Interestingly, those components almost had no effectiveness on the concentration of white blood cells in peripheral blood and bone marrow, which were decreased during CTX therapy. These results suggest that jujube powder displays its function in the way of prebiotics. Taken together, our experimental results confirmed that jujube powder improved CTX efficiency by enhancing infiltration of CD8^+^ T cells into tumor microenvironment, meanwhile suppressing eosinophilia. It should, however, be noted that the current work was performed in a mouse model. Considering the interspecies and individual diversities, the results may be not fully applicable to humans, which needs further study in clinical trials.

## 5. Conclusions

We applied jujube powder to the CTX treatment of MC38 marine colon tumor. We observed that jujube powder raised the diversity of gut microbiota. Especially, the abundance of *Bifidobacteriales* was conductive to the production of SCFAs in the cecum content. Moreover, the administration of jujube powder recovered the concentration of white blood cells in the CTX treatment, among which the percentage of CD8^+^ T cells in peripheral blood was significantly enhanced. The higher level of CD8^+^ T cells among tumor microenvironment is responsible for the enhanced efficacy of CTX, as indicated by the inhibition of the tumor growth. Meanwhile, the administration of jujube powder inhibited the growth of eosinophilia. All these results suggest the promising utilization of jujube powder as prebiotics to regulate gut microbiota for better cancer chemotherapy.

## Figures and Tables

**Figure 1 nutrients-13-02700-f001:**
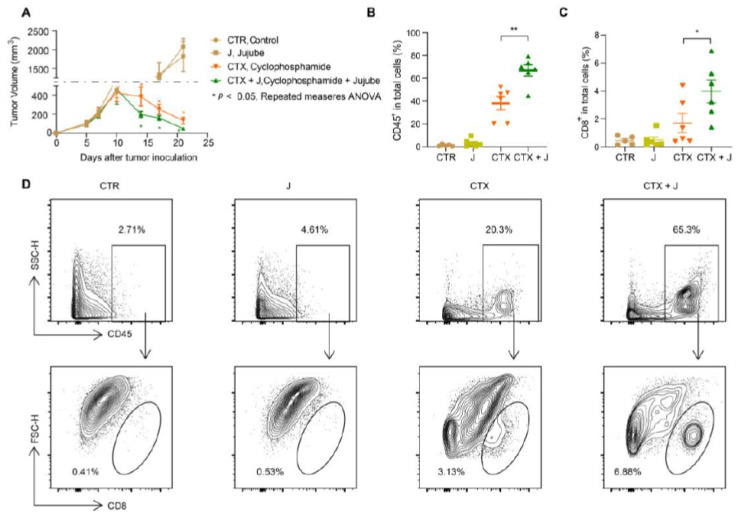
Jujube increased the antitumor efficacy of CTX in mice MC38 colon tumor model. (**A**) All mice were subcutaneously inoculated with MC38 colon cancer cells (about 1 × 10^6^) and i.p. injection of 80 mg/kg CTX dissolved in sterile saline or equal volumes sterile saline on days 7–9. For mice in the J and CTX + J groups, 800 mg/kg jujube powder dissolved in sterile deionized water was admitted daily by gavage from day 10. (**B**) The ration of infiltration of CD45^+^ leukocytes in tumor microenvironment. (**C**) The ration of CD8^+^ T cells of total cells in tumor microenvironment. (**D**) Representative flow cytometry results showing the ration of infiltration CD45^+^ leukocytes and CD8^+^ T cells in the tumor from mice of CTR, J, CTX, or CTX + J group. * *p* < 0.05, ** *p* < 0.01.

**Figure 2 nutrients-13-02700-f002:**
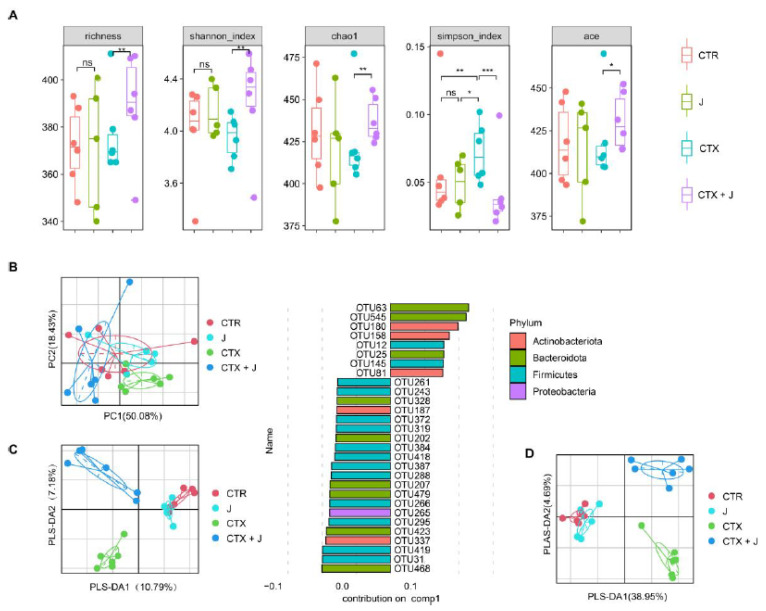
Varieties in alpha and beta diversity indices of gut microbiota with different treatments. (**A**) Alpha diversity of the samples was measured by observed species, Shannon, chao1, Simpson, and ace indices. (**B**) Principal component analysis based on OTU abundance. (**C**) Partial least squared–discriminative analysis plot displayed a distinct separation between CTR, J, CTX, and CTX + J group using all OTUs and the key OTUs contributions to PLS component 1; the colors represent the phylum of the indicated OTU. (**D**) The associated contribution plot demonstrating the top 5% OTUs that influence the PLS–DA. * *p* < 0.05, ** *p* < 0.01, *** *p* < 0.001.

**Figure 3 nutrients-13-02700-f003:**
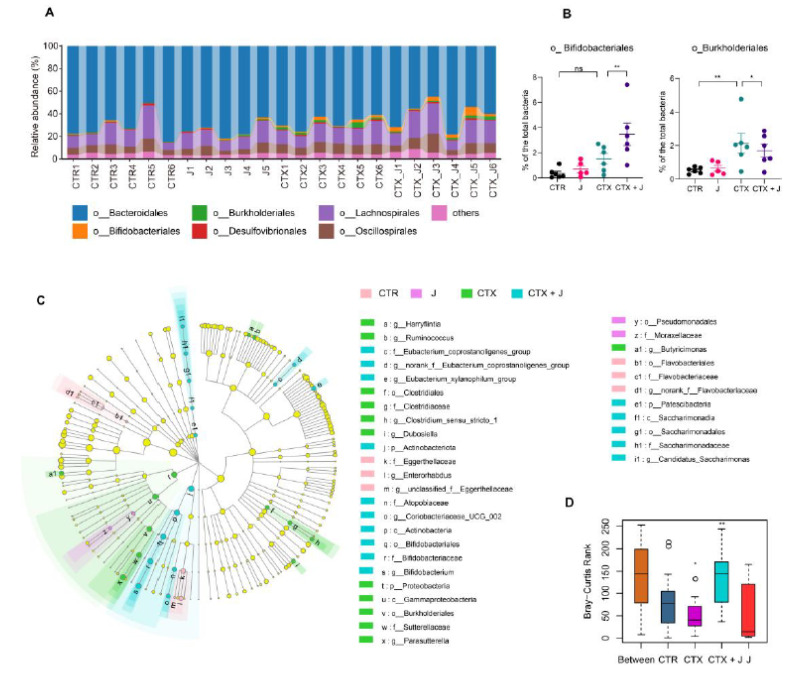
Varieties of gut microbiota composition at different groups. (**A**) The composition of representative bacterial in each group at order level. (**B**) The *Bifidobacteriales* and *Burkholderiales* ratio of the total bacterial. (**C**) The LEfSe analysis of 16S sequences of CTR, J, CTX, and CTX + J group. (**D**) Anosim analysis results of CTR, J, CTX, and CTX + J group. Between represents the difference between groups, others are within groups. The greater the distance is, the greater the difference is. * *p* < 0.05, ** *p* < 0.01.

**Figure 4 nutrients-13-02700-f004:**
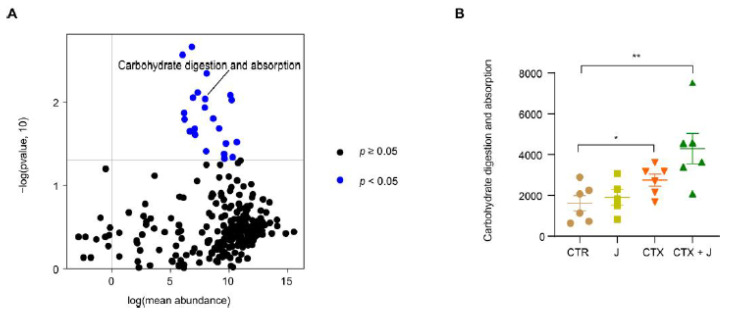
Predictive functional profiling reveals the mechanism of modulation by gut microbiota with jujube. (**A**) PICRUSt analysis cooperates with the Kyoto Encyclopedia of Genes and Genomes (KEGG) database of microbial genomic information; blue points represent significantly various modules between the CTX and CTX + J groups. (**B**) Carbohydrate digestion and absorption was significantly increased in CTX + J group. * *p* < 0.05, ** *p* < 0.01.

**Figure 5 nutrients-13-02700-f005:**
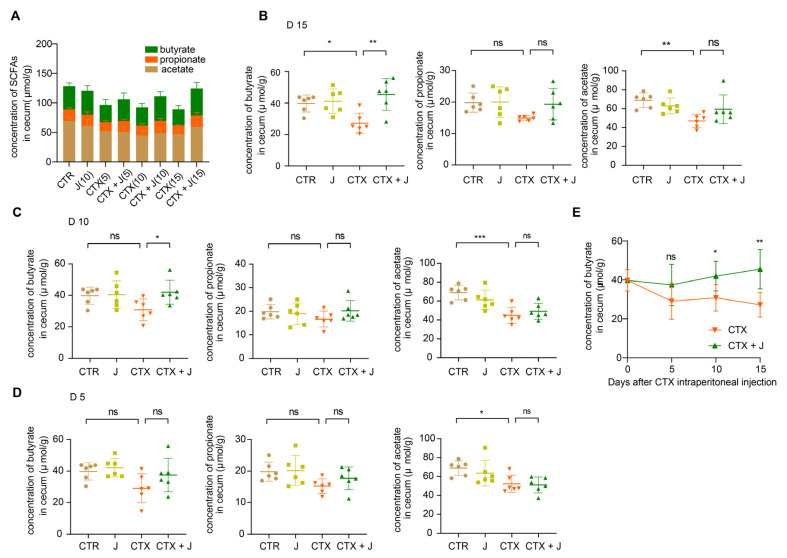
Jujube recover the production of SCFAs in cecum content after CTX treatment. (**A**) The total concentration of SCFAs in cecum content. Quantification of butyrate, propionate, and acetate in cecum content on day 15 (**B**), 10 (**C**), and 5 (**D**) after CTX therapy. (**E**) The trend of butyrate on days 5, 10, and 15. * *p* < 0.05, ** *p* < 0.01, *** *p* < 0.001.

**Figure 6 nutrients-13-02700-f006:**
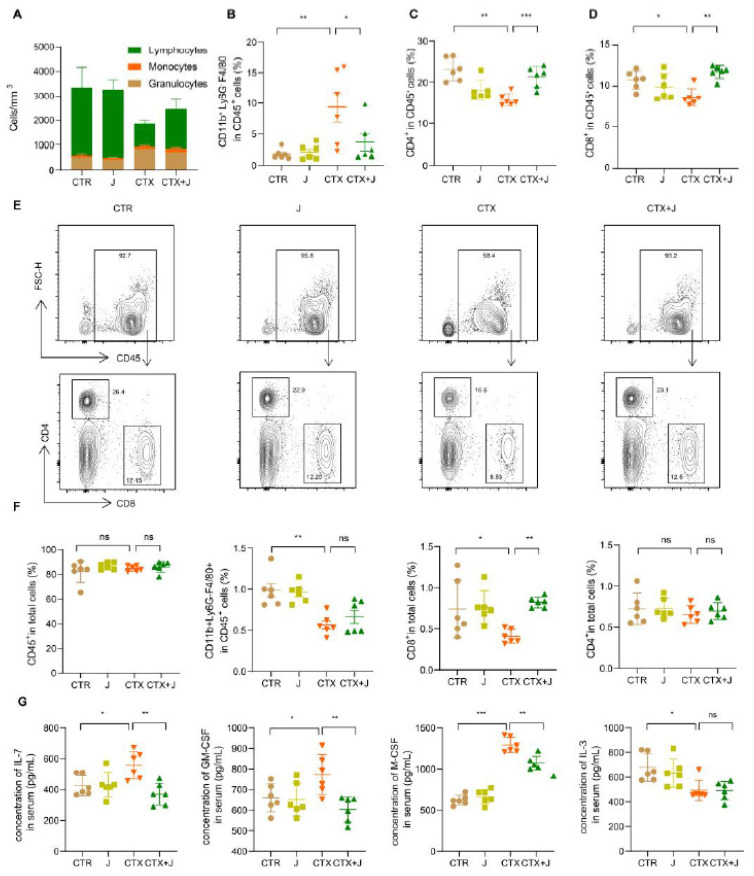
Jujube recuperated the number of Lymphocytes, eosinophils, CD4^+^, and CD8^+^ T cells and regulated the concentration of cytokines in peripheral blood after CTX therapy. (**A**) Quantification of total white blood cells in peripheral blood in each group. (**B**) The ratio CD11b^+^ Ly6G^−^F4/80^+^ cells, CD45^+^CD4^+^ T cells (**C**), and CD45^+^CD8^+^ T cells (**D**) of CD45^+^ cells in peripheral blood. (**E**) Flow cytometry plots of CD45^+^ T cells, CD45^+^CD4^+^ T cells, and CD45^+^CD8_+_ T cells. (**F**) The ratio CD45^+^ cells, CD11b^+^ Ly6G^−^F4/80^+^ cells, CD45^+^CD4^+^ T cells, and CD45^+^CD8^+^ T cells of total cells in bone marrow. (**G**) The concentrations of IL-7, GM-CSF, M-CSF, and IL-3 in serum. * *p* < 0.05, ** *p* < 0.01, *** *p* < 0.001.

**Table 1 nutrients-13-02700-t001:** Cell-marker schemes used for flow cytometry identification of cell population.

Cells	Markers
Total T cells	CD45^+^
CD4^+^ T cells	CD45^+^CD4^+^CD8^−^
CD8^+^ T cells	CD45^+^CD8^+^CD4^−^
B cells	CD19^+^
Myeloid	CD11b^+^
Eosinophils	CD11b^+^Ly6G^−^F4/80^+^SSCh
Monocytes	CD11b^+^Ly6C^+^ Ly6G^−^SSCl
Macrophages	CD11b^+^Ly6C^−^Ly6G^−^SSCl
Granulocytes	CD11b^+^Ly6G^+^SSCm

**Table 2 nutrients-13-02700-t002:** Cell-marker schemes and fluorophore conjugates used for flow cytometry.

Markers	Conjugate	Manufacturer
CD45	APCCy7	BioLegend
CD4	PECy7	BioLegend
CD8a	BrilliantViolet605	BioLegend
CD19	AlexaFluor700	BioLegend
Ly6C	PE	BioLegend
Ly6G	FITC	BioLegend
F4/80	BrilliantViolet421	BioLegend

## Data Availability

The 16S rRNA gene amplicon data and their raw sequence reads are acquirable through the SRA with project ID PRJNA742472. (V. 7.1, http://drive5.com/uparse/, accessed on 29 June 2021).

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
