# Peer review of "Jujube Powder Enhances Cyclophosphamide Efficiency against Murine Colon Cancer by Enriching CD8+ T Cells While Inhibiting Eosinophilia"

_nutrients, 2021, doi:10.3390/nu13082700_

Round 1

Reviewer 1 Report

Dear Author,

The author by Huiren et al., titled "Jujube Powder Enhances Cyclophosphamide Efficiency against Murine Colon Cancer by Enriching CD8+ T Cells while Inhibiting Eosinophilia" was described well. They were trying to show how to  modulate gut microbiota for an enhanced chemotherapy using jujube powder as prebiotics.

The author used R package DESeq2  for bioinformatics analysis.

Jujube powder has better response in the gut microbiota significantly by recovery of the alpha diversity index, increased CD8+ T cells in peripheral blood and tumor microenvironment and inhibition of  eosinophilia. However findings confirmed that the intraperitoneal injection of CTX caused both the reduction of CD8+ T cells and the increase of eosinophil in peripheral blood. Translational study will show CTX efficiency by enhancing infiltration of CD8+T cell into tumor microenvironment.

The manuscript was described well. I suggest, please follow the instruction by the editor.

Thanks

Author Response

Response to Reviewer #1

Point 1: The author by Huiren et al., titled "Jujube Powder Enhances Cyclophosphamide Efficiency against Murine Colon Cancer by Enriching CD8+ T Cells while Inhibiting Eosinophilia" was described well. They were trying to show how to modulate gut microbiota for an enhanced chemotherapy using jujube powder as prebiotics.

Response: The authors thank the reviewer for the appreciation of the significance of this work, and moreover, for the constructive comments.

Point 2: Jujube powder has better response in the gut microbiota significantly by recovery of the alpha diversity index, increased CD8+ T cells in peripheral blood and tumor microenvironment and inhibition of eosinophilia. However, findings confirmed that the intraperitoneal injection of CTX caused both the reduction of CD8+ T cells and the increase of eosinophil in peripheral blood. Translational study will show CTX efficiency.

Response: Thanks for these insightful comments. The reported experimental findings on the reduction of CD8+ T cells and the increase of eosinophil in peripheral blood, and enhanced infiltration of CD8+ T cells infiltration in tumors caused by the injection of CTX have been added in the revised version, as follows.

“Nurrochmad et al observed that CD8+ T cells in peripheral blood were reduced by CTX treatment (150 and 100 mg/kg body weight on day 1 and 4) [31]. Escudero-Vilaplana reported that CTX caused increase of eosinophil [32]. Wu et al demonstrated that CTX treatment (140 mg/kg every 6 days) induced reduction of CD8+ T cells in blood while increased CD8+ T cells infiltration in tumors.[33]”

 (Discussion, Page 10, Line 304-308)

  1. Nurrochmad, A.; Ikawati, M.; Sari, I.P.; Murwanti, R.; Nugroho, A.E. Immunomodulatory Effects of Ethanolic Extract of Thyphonium flagelliforme (Lodd) Blume in Rats Induced by Cyclophosphamide. Journal of Evidence-Based Integrative Medicine 2015, 20, 167-172, doi:10.1177/2156587214568347.
  2. Escudero-Vilaplana, V.; Osorio-Prendes, S.; Collado-Borrell, R.; Gonzalez-Arias, E.; Sanjurjo-Saez, M. Eosinophilia secondary to lenalidomide therapy. Journal of Clinical Pharmacy and Therapeutics 2018, 43, 273-275, doi:10.1111/jcpt.12611.
  3. Wu, J.; Waxman, D.J. Metronomic cyclophosphamide eradicates large implanted GL261 gliomas by activating antitumor Cd8(+) T-cell responses and immune memory. Oncoimmunology 2015, 4, doi:10.1080/2162402x.2015.1005521.

Reviewer 2 Report

The authors presented a valuable study where the jujube powder was used against cancer cells. The paper demonstrates relevant data; however, there are some remarks to revise: 

  • there are some lacking spaces or dots; correct carefully
  • What was the number of animals per group per one experimental condition?
  • what medium was used for MC38? add more details regardring cells harvesting
  • Section 2.5. requires more details, how blood was taken from animals and how were they prepared. How was blood prepared for analysis. 
  • line 155 - here appears not enumerated section "Staining" , all stains are listed in the tables, but there is no protocol for staining; provide more details. 

Author Response

Response to Reviewer #2

Point 1: There are some lacking spaces or dots; correct carefully

Response: Thanks for reviewer’s great efforts. All lacking spaces and dots have been corrected in the revised version.

Point 2: What was the number of animals per group per one experimental condition?

Response: Each group has 6 mice per one experimental condition except for bioinformatics analysis. The jujube group has 5 mice because the concentration of one mouse’s fecal DNA do not conform to the requirement of 16S rRNA gene sequencing. Above information has been added in the revised version.

(Materials and Methods, Page 2, Line 95-98)

Point 3: what medium was used for MC38? add more details regarding cells harvesting

Response: Thanks. Above information has been added to the revised version as follows,

“MC38 from American Type Culture Collection (ATCC) was cultured in 5% CO2 at 37 ℃ using Dulbecco’s Modified Eagle’s Medium (Invitrogen, Carlsbad, CA, USA) with addition of 10% heat-inactivated fetal bovine serum (HyClone, Logan,UT,USA). Then MC38 cells were digested to single cells by 5 minutes treatment with 0.5% trypsin (Sigma-Aldrich, USA). The digestion was terminated by adding Dulbecco’s Modified Eagle’s Medium and subjecting to 5 minutes centrifugation at 400 × g, 4 ℃. The concentrated was suspended to 1 × 107 /ml in phosphate buffered saline (PBS) for subcutaneous injection.”

(Materials and Methods, Page 3, Line 103-110)

Point 4: Section 2.5. requires more details, how blood was taken from animals and how were they prepared. How was blood prepared for analysis.

Response: Thanks for this instruction. Section 2.5 has been revised as follows, “Mice were bled by retro-orbital sinus puncture into heparinized capillary tubes at 5, 10, 15 days of CTX treatment and analyzed using XN-1000V automated system (Sysmex).”

(Materials and Methods, Page 3, Line 140)

Point 5: Line 155 - here appears not enumerated section "Staining", all stains are listed in the tables, but there is no protocol for staining; provide more details.

Response 5: Thanks for this instruction. The staining protocol has been added in the revised version as follows.

“Cells were firstly bound to anti-CD16/32 antibody and then stained with antibody mixes (CD45, CD4, CD8a, Ly6C, Ly6G, Cd11b, F4/80, CD19) at concentration 1:200 for each surface marker antibody in Hank’s buffered saline solution for 15 minutes in dark environment. All samples were washed with 1 mL Hank’s buffered saline solution and centrifuged for 5 minutes (400 × g, 4 ℃). The cells were suspended in the Hank’s buffered saline solution and applied to LSRII cell analyzers (BD Biosciences). The multicolor flow data analysis was performed using FlowJo 10.6.2.”

(Materials and Methods, Page 3, Line 164-170)
